# EXPANDING EXPRESSIVITY IN TRANSFORMER MODELS WITH MÖBIUSATTENTION

## ABSTRACT

Attention mechanisms and Transformer architectures have revolutionized Natural Language Processing (NLP) by enabling exceptional modeling of long-range dependencies and capturing intricate linguistic patterns. However, their inherent reliance on linear operations in the form of matrix multiplications limits their ability to fully capture inter-token relationships on their own. We propose MöbiusAttention, a novel approach that integrates Möbius transformations within the attention mechanism of Transformer-based models. Möbius transformations are non-linear operations in spaces over complex numbers with the ability to map between various geometries. By incorporating these properties, MöbiusAttention empowers models to learn more intricate geometric relationships between tokens and capture a wider range of information through complex-valued weight vectors. We build and pre-train a BERT and a RoFormer version enhanced with MöbiusAttention, which we then finetune on the GLUE benchmark. We evaluate empirically our approach against the baseline BERT and RoFormer models on a range of downstream tasks. Our approach compares favorably against the baseline models, even with smaller number of parameters suggesting the enhanced expressivity of MöbiusAttention. This research paves the way for exploring the potential of Möbius transformations in the complex projective space to enhance the expressivity and performance of foundation models.

## 1 INTRODUCTION

Transformers (Vaswani et al., 2017) have revolutionized various areas of machine learning, becoming the foundation for groundbreaking models in Natural Language Processing (NLP) like text generation (GPT3 (Brown et al., 2020), BERT (Devlin et al., 2019), Mistral 7B(Jiang et al., 2023)) and computer vision (ViT (Dosovitskiy et al., 2021) utilized in SAM (Kirillov et al., 2023), DINO (Caron et al., 2021) and the multi-modal CLIP (Radford et al., 2021)). At the heart of their success lies the attention mechanism (Vaswani et al., 2017), a powerful tool that enables them to identify relationships between different parts of the data, be it words in a sentence or image patches in a scene.

Despite their remarkable impact, current transformers face limitations. A key constraint is the inherent linearity of the attention mechanism, which primarily relies on weights learned through linear transformations, matrix multiplications, and the softmax function. While softmax is a non-linear operation, it is only used to produce a probability distribution over the elements signaling their relative importance in comparison to the others, and not to introduce non-linear interdependencies. Predominantly linear operations restrict the ability of models to capture complex linguistic dependencies, leading to potential information loss within each attention layer as shown by recent research (Zhang, 2023). Simply increasing the depth of the architecture does not fully solve this issue as it has drawbacks: a) it yields diminishing returns or redundancy, b) it results in considerable computational overhead, offering only partial resolution to the issue (Lee et al., 2021; Stock, 2021), and c) while deeper architectures alleviate information loss to some degree, the accumulation of layers may lead to the re-learning of similar information, possibly causing overfitting.

Therefore, introducing non-linearity directly within the attention mechanism seems to be beneficial. Existing approaches like RoPe (Su et al., 2024) explore this avenue through rotating the query and key vectors based on token positions, while Neural Attention (Zhang, 2023) employ multi-layer perceptrons (MLPs) with non-linear activations to learn the query, key, and value weights $(Q, K, V)$.

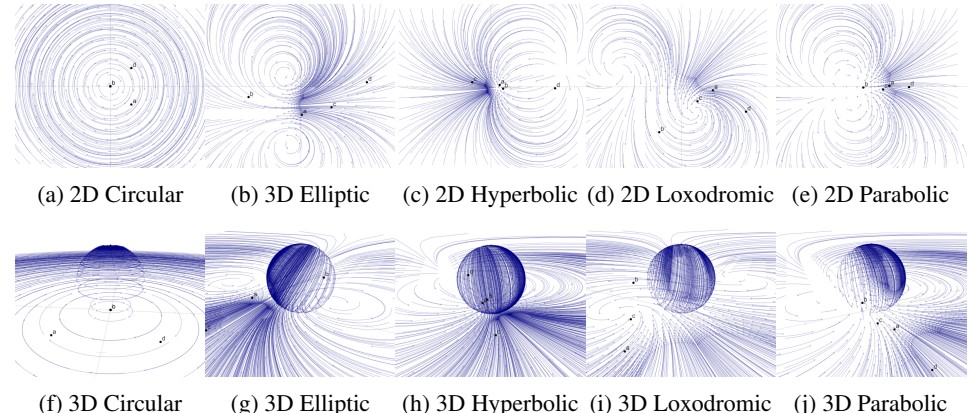

(a) 2D Circular    (b) 3D Elliptic    (c) 2D Hyperbolic    (d) 2D Loxodromic    (e) 2D Parabolic

(f) 3D Circular    (g) 3D Elliptic    (h) 3D Hyperbolic    (i) 3D Loxodromic    (j) 3D Parabolic

Figure 1: Various Möbius transformations: Each sub-figure shows flows from a single point after successive transformations. **Circular** Möbius forms circles. **Elliptic** Möbius has two fixed points at the centers of two circular flows. **Hyperbolic** Möbius flows start from one fixed point and end at another. **Loxodromic** Möbius features spiraling flows between a source and a sink fixed point. **Parabolic** Möbius has two sets of circular flows converging at a single fixed point. 3D visualizations show flows from the Complex plane projected onto the Riemann sphere for Möbius transformations[1].

However, the existing body of research does not explore transformations capable of operating across diverse geometric spaces. Within the attention mechanism, the $Q, K$ and $V$ vectors collectively encode the "necessity," "nature," and "contribution" of each token, respectively. This interplay facilitates the derivation of a token's importance within a sequence and its relationship to other tokens. Consequently, advancements in methods for learning these weights $(Q, K, V)$ have the potential to yield significant performance improvements. However, none of the aforementioned works alter the structure of the weights, not capitalizing on this opportunity.

To address this gap, our work proposes a novel approach called MöbiusAttention. We introduce non-linearity into the attention mechanism through Möbius transformations. These transformations are advantageous because they can map points between different geometries, such as from a line to a circle, a circle to a line, and similarly among lines and circles. Moreover, they encompass various geometric shapes, including Circular, Elliptic, Parabolic, Hyperbolic, and Loxodromic forms, illustrated in Figure 1. These properties allow the model to capture more complex inter-token dependencies than traditional linear methods, which is essential for effective NLP tasks and beyond.

We show that MöbiusAttention can be easily integrated into Transformer-based models, either replacing or in combination with standard attention mechanisms. This integration leads to improved performance across various NLP tasks without necessarily increasing the model size. Specifically, we implement MöbiusAttention-enhanced BERT (Devlin et al., 2019) and RoFormer (Su et al., 2024) models, pre-trained on the C4 dataset (Raffel et al., 2020) and fine-tuned on the GLUE benchmark (Wang et al., 2018). Our evaluations show that our models surpass the performance of the baselines across a suite of tasks designed to assess a model's ability to understand complex linguistic relationships.

## 2 RELATED WORK

### 2.1 REVISITING ATTENTION

The landscape of attention mechanism research is rich and multifaceted, with various approaches aiming to improve different aspects.

A significant portion of research focuses on enhancing the time and memory efficiency of attention, e.g., HyperAttention (Han et al., 2024), FlashAttention (Dao et al., 2022) and other notable works

---

[1]We used the visualization tool in https://timhutton.github.io/mobius-transforms/ to get our visualizations.

(Shen et al., 2021; Child et al., 2019). These works prioritize maintaining functional and mathematical equivalence to the standard attention mechanism.

Several works explore incorporating non-linear functions within the attention algorithm. Linear Transformers (Katharopoulos et al., 2020), Skyformer (Chen et al., 2021), Performers(Choromanski et al., 2021), Cosformer (Qin et al., 2022) and Kerformer (Gan et al., 2023) propose attention mechanisms with reduced computational requirements and comparable or better performance. These approaches utilize non-linear kernels on the learned weight vectors, essentially replacing the softmax function within attention. In comparison, our approach targets the weight representation and learning process itself to enhance its information capture capabilities.

Several approaches introduce non-linear kernels to replace the standard dot-product similarity operation on the $Q$ and $K$ vectors (e.g., Rymarczyk et al. (2021); Tsai et al. (2019); Kim et al. (2019)). In contrast, our approach focuses on modifying the weight representation during the learning process itself, intending to facilitate the acquisition of information-richer vectors.

RoPE (Su et al., 2024) and NeuralAttention (Zhang, 2023) exhibit the most similarity to our work. Both papers introduce non-linear transformations on the $Q, K, V$ weights vectors through rotation or learning via a non-linear MLP activation. However, these methods are limited to mapping within a single geometric space, lacking the flexibility to handle diverse geometries like elliptic, circular, or loxodromic, crucial for capturing intricate inter-token relationships. Furthermore, while both methods operate in real space, our approach leverages the complex domain and operations naturally supported by complex numbers, facilitating the modeling of various phenomena, including cyclical patterns.

### 2.2 COMPLEX-VALUED TRANSFORMER

The exploration of complex-valued models has gained significant traction in recent years, with applications emerging across various domains (Vasudeva et al., 2022; Li et al., 2020; Barrachina et al., 2021; Trabelsi et al., 2018; Nayyeri et al., 2021; Azizi et al., 2022). While complex-valued Transformers have been proposed (Complex Transformer (Yang et al., 2020), Signal Transformer (Peng et al., 2024) and $\mathbb{C}$-Transformer (Eilers & Jiang, 2023)), these works primarily focus on the signal processing field and aim for a complete adaptation of the Transformer architecture to the complex domain, without introducing any alterations to the attention mechanism that are not necessary for the transition from real to complex space. Our work delves deeper into the core component of Transformers, the attention mechanism, seeking novel enhancements.

A complex-valued Transformer specifically for NLP is developed by Wang et al. (Wang et al., 2020) where they define word embeddings as continuous functions over the position of the words in the complex domain, allowing for word representations to change gradually as positions increase. While our work shares the goal of capturing ordered relationships (a facet of geometric properties), we employ a distinct strategy by leveraging transformations which can represent a very broad variation of behavior based on only few learnable parameters. Additionally, our work adopts a new approach to position embeddings, as we use token embeddings as the real part of the input into the model, and the corresponding position embeddings - as the imaginary one. This targeted focus on position embeddings differentiates our approach from existing works.

## 3 BACKGROUND

This section presents all the necessary mathematical background essential for introducing our model.

**Projective Geometry** Salomon (2007); Richter-Gebert (2011); Nayyeri et al. (2021) is a branch of mathematics that studies properties and relationships unaffected by perspective or projection. It focuses on fundamental geometric concepts like points, lines, and planes, considering them as elements of *equivalence classes* rather than distinct entities.

*Coordinate in Projective Geometry* Projective geometry employs *homogeneous coordinates*, representing $N$-dimensional coordinates with $N + 1$ parameters. For instance, a point in 2D Cartesian coordinates, $[X, Y]$, transforms into $[x, y, k]$ in homogeneous coordinates, where $X = x/k$ and $Y = y/k$.

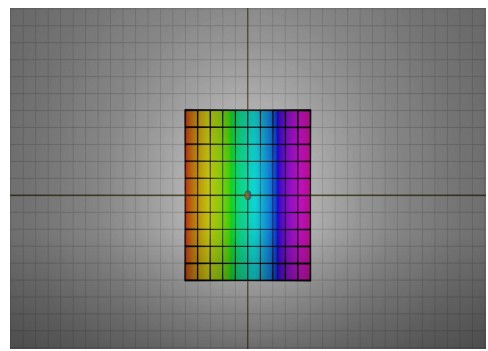 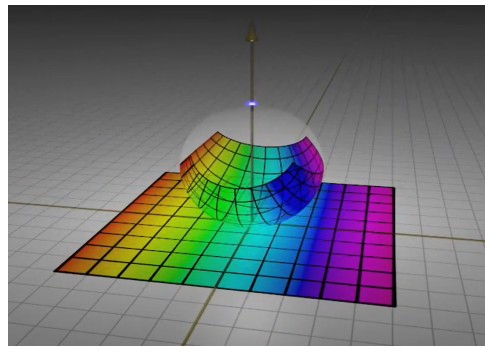

(a) Complex plane with real (x-axis) and imaginary (y-axis) axes. The colorful object is a grid on the Complex plane.

(b) Riemann Sphere positioned on the Complex plane. The grid is projected on the Riemann sphere using stereographic projection.

Figure 2: The Riemann sphere, visualized in Arnold & Rogness (2008), is created by wrapping the Complex plane where the infinite points are projected on the north pole of the sphere. The Riemann sphere is used as a tool for Möbius transformation. A line on the grid is projected as a curve on the sphere.

*Projective Line* A Projective Line serves as the foundational space for projective geometry. To hold the axiom that "two parallel lines intersect at infinity", projective geometry necessitates the inclusion of a point at infinity. Consequently, an extended line $\mathbb{P}^1(\mathbb{K})$ (with $\mathbb{K}$ representing the real line) is constructed, incorporating both $\mathbb{K}$ and a point at infinity, topologically resembling a circle. Formally, the projective line is expressed as the set $\{[x, 1] \in \mathbb{P}^1(\mathbb{K}) | x \in \mathbb{K}\}$, augmented by an additional element $[1 : 0]$ representing the point at infinity. In this paper, we are interested in the *Complex projective line* denoted by $\mathbb{CP}^1$ ($\mathbb{K} = \mathbb{C}$) due to its favorable geometric properties introduced in the subsequent sections.

*Riemann Sphere* [2] The Riemann Sphere, depicted in Figure 2b, extends the concept of the complex plane (Figure 2a) by including a point at infinity. It is constructed by mapping the points on the complex plane onto a sphere by using the stereographic projection, where poles represent 0 and $\infty$. In projective geometry, the Riemann Sphere serves as a complex projective line, offering valuable insights for projective transformations.

*Projective and Möbius Transformations* Salomon (2007); Richter-Gebert (2011); Nayyeri et al. (2021) A Projective Transformation involves mapping the Riemann Sphere onto itself. Suppose $[x : y] \in \mathbb{CP}^1$ be a point in the Complex projective line, represented in the homogeneous coordinates. A projective transformation in $\mathbb{CP}^1$ can be denoted by a mapping $\mathcal{T} : \mathbb{CP}^1 \to \mathbb{CP}^1$ which is a matrix-vector multiplication:

$$\mathcal{T}([x, y]) = \mathbf{M} \begin{bmatrix} x \\ y \end{bmatrix}, \quad \mathbf{M} = \begin{bmatrix} a & b \\ c & d \end{bmatrix}, \tag{1}$$

where the matrix $\mathbf{M}$ must be invertible ($\det(\mathbf{M}) \neq 0$). Identifying $\mathbb{CP}^1$ with $\hat{\mathbb{C}} = \mathbb{C} \cup \infty$, a projective transformation is represented by a fractional expression through a sequence of bringing a point in the complex plane to homogeneous coordinate, applying a transformation, and bringing back from homogeneous coordinate to the Complex space as:

$$x \to [x\ 1] \to \begin{bmatrix} a & b \\ c & d \end{bmatrix} \begin{bmatrix} x \\ 1 \end{bmatrix} \to \begin{bmatrix} ax + b \\ cx + d \end{bmatrix} \to \frac{ax + b}{cx + d}, \tag{2}$$

where the mapping $\mathcal{M} : \hat{\mathbb{C}} \to \hat{\mathbb{C}}$ is the Möbius transformation defined as:

$$\mathcal{M}(x) = \frac{ax + b}{cx + d}, \quad ad - bc \neq 0, \ a, b, c, d, x \in \mathbb{C}. \tag{3}$$

---

[2]we refer to https://www.youtube.com/watch?v=0z1fIsUNhO4&t=32s for detailed explanation of Möbius transformation

| Function | Parabolic | Circular | Elliptic | Hyperbolic | Loxodromic |
|---|---|---|---|---|---|
| Condition | $tr\mathbf{M}^2 = 4$ $(\Delta = 0)$ | $tr\mathbf{M}^2 = 0$ $(\Delta = 0)$ | $0 < tr\mathbf{M}^2 < 4$ $(\Delta < 0)$ | $tr\mathbf{M}^2 > 4$ $(\Delta > 0)$ | $tr\mathbf{M}^2 \notin [0,4]$ $(\Delta > 0)$ |
| Isomorphic | $\begin{bmatrix} 1 & a \\ 0 & 1 \end{bmatrix}$ | $\begin{bmatrix} i & 0 \\ 0 & -i \end{bmatrix}$ | $\begin{bmatrix} e^{i\theta/2} & 0 \\ 0 & e^{-i\theta/2} \end{bmatrix}$ | $\begin{bmatrix} e^{\theta/2} & 0 \\ 0 & e^{-\theta/2} \end{bmatrix}$ | $\begin{bmatrix} k & 0 \\ 0 & \frac{1}{k} \end{bmatrix}$ |

Table 1: Types of Möbius transformations and their conditions. *tr* denotes the trace of a matrix.

*Möbius Group* The Möbius Group comprises all Möbius transformations, forming the projective linear group $PGL(2, \mathbb{C})$. It consists of all invertible $2 \times 2$ matrices with matrix multiplication as the operation in a projective space. Denoted by $Aut(\hat{\mathbb{C}})$, it serves as the automorphism group of the Riemann sphere $\hat{\mathbb{C}}$, or equivalently, $\mathbb{CP}^1$. Due to the isomorphism between the projective linear group $PGL(2, \mathbb{C})$ and the Möbius group, denoted as $PGL(2, \mathbb{C}) \cong Aut(\hat{\mathbb{C}})$ (Kisil, 2012), the characteristics specified for Equation 3 also hold true for Equation 1.

*Variants Of Möbius Transformation* Each Möbius transformation yields a maximum of two fixed points, $\gamma_1$ and $\gamma_2$, on the Riemann sphere, determined by solving $\mathcal{M}(\gamma) = \gamma$ (Richter-Gebert, 2011)

$$\gamma_{1,2} = \frac{(a - d) \pm \sqrt{\Delta}}{2c}, \tag{4}$$

where $\Delta = (tr\mathbf{M})^2 - 4 \det \mathbf{M}$. Based on the number of fixed points, Möbius transformations are categorized into Parabolic or Circular (one fixed point), Elliptic, Hyperbolic, and Loxodromic (two fixed points) transformation functions. Refer to Figure 1 and Table 1 for detailed conditions. Each group of transformations forms a subgroup that is isomorphic to the group of matrices listed under the *Isomorphic* row in Table 1.

Each transformation possesses a characteristic constant $k = e^{\alpha + i\beta}$, signifying the *sparsity/density* of the transformation. The parameter $\beta$ represents the expansion factor, delineating the repulsive nature of the fixed point $\gamma_1$ and the attractive quality of the second fixed point $\gamma_2$. Meanwhile, $\alpha$ serves as the rotation factor, dictating the extent to which a transformation rotates the plane counterclockwise around $\gamma_1$ and clockwise around $\gamma_2$.

## 4 MÖBIUS ATTENTION

In this section, we introduce our novel attention mechanism centered around the Möbius transformation. We present our approach through the following components: a) *token and position representation*, b) *query, key, and value computation*, and c) *attention calculation*.

**Token and Position Representation**  Let $\mathbb{T} = \{w_i\}_{i=1}^N$ be a set of $N$ input tokens. Each token $w_i$ has a position in a text, denoted by $p_{w_i}$. Each token $w_i$ and its position $p_{w_i}$ are embedded as a $d$ dimensional real vector, denoted by $\mathbf{w}_i, \mathbf{p}_{w_i} \in \mathbb{R}^d$. A pair token-position $\rho_i = (w_i, p_{w_i})$ is represented as a $d$ dimensional complex number, i.e., $\boldsymbol{\rho}_i = \mathbf{w}_i + i\mathbf{p}_{w_i} \in \mathbb{C}^d$. Thus, each element of $\boldsymbol{\rho}_{ij}, j = 1, \ldots, d$ is a point in the Complex plane (Figure 2a), i.e., $\boldsymbol{\rho}_{ij} \in \mathbb{C}$.

**Query Representation**  Each element of the attention matrix, e.g., $a_{uv}$, determines the similarity between the source token $w_u$ and the target token $w_v$. The vanilla transformer defines the query $q(x)$ and the key functions $k(x)$ as a linear function. In contrast, we propose to define the query function as an element-wise Möbius transformation. We present the query function in two equivalent representations:

*Möbius Query Representation* We define the Möbius query function $\mathcal{M}_q(x) = [\mathcal{M}_{q_1}, \ldots, \mathcal{M}_{q_d}] \in \mathbb{C}^d$ as follows

$$\mathcal{M}_{q_j}(\boldsymbol{\rho}_{ij}) = \frac{a_{q_j}\boldsymbol{\rho}_{ij} + b_{q_j}}{c_{q_j}\boldsymbol{\rho}_{ij} + d_{q_j}}, j = 1, \ldots, d, \tag{5}$$

where $a_{q_j}, b_{q_j}, c_{q_j}, d_{q_j} \in \mathbb{C}$. Because $PGL(2,\mathbb{C}) \cong Aut(\hat{\mathbb{C}})$, we can present the projective representation of the query function as follows.

*Projective Query Representation* To gain better insight and model interpretation, we introduce the projective representation of the query function, denoted as $\mathcal{T}_q(x) = [\mathcal{T}q_1(x), \ldots, \mathcal{T}_{q_{d(x)}}]$. To apply the projective transformation, we first bring the pair token-position representation $\boldsymbol{\rho}_i$ into a homogeneous coordinate, represented by $\boldsymbol{\rho_i^h} \in \mathbb{CP}^d$. With this, the projective query function is defined as follows:

$$\mathcal{T}_{q_j}(\boldsymbol{\rho}_{ij}^h) = \boldsymbol{M}_{q_j}\boldsymbol{\rho}_{ij}^h, j = 1, \ldots, d, \tag{6}$$

where $\boldsymbol{M}_{q_j} = \begin{bmatrix} a_{q_j} & b_{q_j} \\ c_{q_j} & d_{q_j} \end{bmatrix}$. The Equation 6 shows that the query calculation can be done in matrix-vector products in the projective space, enabling efficient implementation through tensor products.

**Key and Value Representation**  A complex linear transformation is used for key $\mathcal{K}(.)$ and value $\mathcal{V}(.)$ functions as follows

$$\mathcal{K}_j(\boldsymbol{\rho}_{ij}) = \mathbf{w}_{kj}\boldsymbol{\rho}_{ij}, j = 1, \ldots, d, \tag{7}$$
$$\mathcal{V}_j(\boldsymbol{\rho}_{ij}) = \mathbf{w}_{vj}\boldsymbol{\rho}_{ij}, j = 1, \ldots, d. \tag{8}$$

**Möbius Attention**  Similar to Vaswani et al. (2017), we compute the Möbius attention as follows

$$Att(\mathcal{T}_q, \mathcal{K}, \mathcal{V}) = softmax(\frac{\mathcal{O}}{\sqrt{d}})\mathcal{V}, \tag{9}$$

where $\mathcal{O}$ is the attention matrix. Defining the Complex query matrix $\boldsymbol{Q}$ and the key matrix $\boldsymbol{K}$ with elements $\boldsymbol{Q}_{ij} = \mathcal{T}_{q_j}(\boldsymbol{\rho_{ij}}), \boldsymbol{K}_{ij} = \mathcal{K}_j(\boldsymbol{\rho_{ij}}), i = 1, \ldots, N, j = 1, \ldots, d$, we compute the matrix $\mathcal{O}$ by $\boldsymbol{Q}\boldsymbol{K^T}$. The rest of the architecture is similar to the one introduced in Vaswani et al. (2017). In this paper, we integrate the Möbius attention in the BERT model (Devlin et al., 2019), and into RoFormer (Su et al., 2024), essentially a BERT model with rotary positional embeddings (RoPe). The detailed architecture is presented in the experiments section.

**Geometric Interpretation**  In this section, we provide a geometric interpretation of our attention model and highlight its advantages compared to existing models.

*Capturing Local Information* The set of all query matrices $\boldsymbol{M}_{q_j}$ in Equation 6 constitutes the generalized linear group $GL(2,\mathbb{C})$. If we impose the condition $\det \boldsymbol{M}_{q_j} = 1$, we obtain the special linear group $SL(2,\mathbb{C})$, which preserves both volume and orientation. Consequently, the set of source token-position pairs in a sequence can be mapped to the set of target token-position pairs, capturing local dependencies between tokens within the attention matrix.

*Capturing Global Information* When $\det \boldsymbol{M}_{q_j} \neq 1$, the transformation alters both volume and orientation. This property, combined with the Möbius transformation's capability to map lines to circles and vice versa, results in a more expressive attention matrix. This enhanced expressiveness captures more intricate relationships between tokens and understands complex linguistic patterns.

In more detail, Möbius transformations offer a robust framework for analyzing and interpreting text. By leveraging these transformations, we can transition between various geometric shapes—lines, and circles—in a manner that preserves the structural integrity of the data. This adaptability is crucial for modeling the nuanced dependencies that exist between different tokens in a sequence.

**Time and Space Complexities**  Despite the favorable characteristics of our model, it is efficient in terms of time and space complexities. The time complexity of Möbius attention is $O(n^2d + nd^2)$ where $d$ is the token vector size and $n$ is the number of tokens in the sequence in the case of self-attention. The space complexity of MöbiusAttention is similar to the vanilla attention $O(n^2)$. We will later show in our experiment that our approach requires fewer layers than the vanilla model and is more efficient in memory and time.

## 5 EXPERIMENTS

**Experimental Setup**    We integrate MöbiusAttention into the BERT and RoFormer architectures (Devlin et al., 2019; Su et al., 2024) using the MosaicBERT framework (Portes et al., 2023), licensed under the Apache 2.0 License, instead of the original BERT framework, which was also used for RoFormer. This choice is motivated by several factors, including its ease of adaptation, extensibility to additional models, and suitability for training on the C4 dataset. See Appendix A.1 for further details on our motivation.

For training, we employ a cluster with four A100-40GB GPUs. The software environment consists of PyTorch version 1.13.1, CUDA version 11.7, Python version 3.10, and Ubuntu version 20.04.

Given that the results reported in Portes et al. (2023) were obtained using a setup of 8 A100-80GB GPUs, while our setup consists of 4 A100-40GB GPUs, we opted to train the baseline ourselves rather than directly adopting their results. Following the specifications of the framework used, we pretrain all models for 70,000 steps with a batch size of 4096.

**Datasets**    In contrast to the BookCorpus (Zhu et al., 2015) and English Wikipedia combination used for pre-training BERT and RoFormer (Devlin et al., 2019; Su et al., 2024), we leverage the more recent and larger Colossal Clean Crawled Corpus (C4) dataset (Raffel et al., 2020), licensed ODC-By, for our pre-training stage. This aligns with the recent trend of training NLP models on increasingly vast datasets, a strategy demonstrably leading to performance improvements witnessed in models succeeding BERT (e.g., Liu et al. (2019); Raffel et al. (2020); Liu et al. (2021); Lee-Thorp et al. (2022)). By adopting the C4 dataset, we not only benefit from this advancement but also ensure consistency with the MosaicBERT framework, which is specifically optimized for this data source.

**Models**    Our study employs two pre-trained transformer models as baselines: BERT (Devlin et al., 2019) and RoFormer (Su et al., 2024). BERT was selected due to its popuarity and frequent adoption as the foundation for numerous high-performing models (e.g., Liu et al. (2019); He et al. (2021); Lan et al. (2019)). RoFormer serves as our second baseline, chosen for its derivation from BERT and its integration of rotary positional embeddings (RoPe). RoPe introduces a geometric dimension to the model by rotating the query and key vectors according to token positions, employing circular geometry. Additionally, RoPe has been integrated in a multitude of novel LLMs such as LLAMA 1, 2 and 3 (Touvron et al., 2023a;b; Dubey et al., 2024), the Falcon series (Almazrouei et al., 2023), PaLM (Chowdhery et al., 2023), GPT-NeoX (Black et al., 2022), etc. Additional details on our motivation for these selections can be found in Appendix A.1.

We use the implementation from the Hugging Face Transformers library for PyTorch[3]. We use the base uncased version without any modifications to the architectures.

We also created our BERT and RoFormer versions enhanced with MöbiusAttention - MöbiusBERT and MobRoFormer. The Möbius transformation boasts high expressivity, but a Transformer solely comprised of MöbiusAttention blocks would likely suffer from overfitting, as we show in our ablation study in Section 5. To address these limitations, we strategically integrate MöbiusAttention - MöbiusBERT and MobRoFormer utilize MöbiusAttention only in the first and the last layer while relying on standard Transformer blocks for the remaining layers. Additionally, we allow for adjustment of the percentage of MöbiusAttention not only on layer-level but also on head-level, so it can range from zero to full utilization. We propose combining MöbiusAttention with vanilla attention within the same layer by introducing an architecture that allows us to set the percentage of heads using MöbiusAttention. We used an equal split of 50% vanilla attention heads (6 heads) and 50% Möbius Attention heads. Other variants with different placements of MöbiusAttention are offered for MöbiusBERT in our ablation study.

Each block utilizing Möbius Attention operates in the Complex space, necessitating several architectural adjustments. To construct the real and imaginary input channels for the first layer, we separate the word and positional encodings of the token-position pairs to represent the real and imaginary components, i.e., $\rho_i = (w_i, p_{w_i})$ gives us the input to the model $I = \{\rho_i\}_{i=1}^N$ with $\boldsymbol{\rho}_i = \mathbf{w}_i + i\mathbf{p}_{w_i} \in \mathbb{C}^d$. This strategy is not applicable for the last block which again uses MöbiusAttention, since it is preceded by vanilla Transformer layers operating in the real space. For this last layer, we build the

---

[3]https://github.com/huggingface/transformers, Accessed: 26.09.2024

complex input by taking the real-valued output of the preceding layer as input to the real channel and the token embeddings from the real channel of the first block, i.e., the other MöbiusAttention block, as input to the imaginary channel. A visualization of the input construction is provided in Fig. 4b in Appendix A.2.

The output of the complex MöbiusAttention layers is converted back to real space by adding the real and imaginary outputs, i.e., for first and last layers $l \in \{1, L\}$ the output of the specific layer is $out_l = (O_{r,l} + O_{i,l})$. For additional details on the architecture and design, please refer to Appendix A.2.

The chosen approach grants a stronger emphasis on positional encodings by separating them into a distinct channel. The key advantage of framing vanilla attention with MöbiusAttention lies in its ability to leverage both the two input channels and the expressive power of Möbius transformations. Furthermore, the presence of two input channels before the final block allows for a residual connection to earlier layers using the imaginary channel without affecting the significance of the previous block's output, which remains in the real channel. Figure 4b illustrates the architecture of MöbiusBERT.

**Low-Dimensional Möbius Models**    The usage of complex-valued parameters and Möbius transformation introduces additional parameters. Accordingly, we reduce the depth of the Möbius models in order to ensure a fair comparison to match the parameter count of the BERT baseline. As an alternative, we also create a low-dimensional MöbiusAttention version which uses less parameters than the original one. Here the Möbius transformation is applied after the linear query transformation, thus lowering the number of parameters.

**Tasks**    To ensure maximal comparability between the models, we adhere to the setup for the pre-training and finetuning of BERT as specified in Devlin et al. (2019). The only deviation on our end is choosing Masked Language Modeling (MLM) as the only pre-training objective, leaving out Next Sentence Prediction (NSP) objective. We pre-train all models with this setup (BERT, RoFormer and the Möbius models). This choice of ours is in correspondence to newer research works showing that the NSP task is obsolete given MLM (Conneau & Lample, 2019; Liu et al., 2019; Joshi et al., 2020; Raffel et al., 2020; Yang et al., 2019; Su et al., 2024). With this exception, the remainder of decisions regarding pre-training are adopted from BERT, and, correspondingly, the setup for BERT-Base in the framework of our choice: masking ratio 15%, and dropout 0.1.

To assess the performance of the pre-trained models for different NLP tasks, we fine-tune and evaluate them on the GLUE benchmark (Wang et al., 2018). Details on individual tasks are in Appendix A.4.

| Model | Layers | Parameters | MNLI-(m/mm) | QQP (Acc/F1) | QNLI | SST-2 | CoLA | RTE | STS-B | MRPC (Acc/F1) | AVG |
|---|---|---|---|---|---|---|---|---|---|---|---|
| BERT (our benchmark) | 12 | 110M | 84.46/85.14 | 91.23/88.13 | 90.65 | 92.16 | 56.29 | 76.61 | 89.79 | 87.40/90.88 | 83.64 |
| RoFormer (our benchmark) | 10 | 110M | 84.17/84.57 | 91.33/88.34 | 90.74 | 92.32 | 53.16 | 77.62 | 89.04 | 89.36/92.3 | 83.49 |
| Overall (others) | / | / | 84.46/85.14 | 91.33/88.34 | 90.74 | 92.32 | 56.29 | 77.62 | **89.79** | **89.36/92.3** | 84.03 |
| MobRoFormer H | 10 | 114M | 84.58/85.03 | 91.36/88.39 | 91.20 | 92.05 | 55.65 | 77.40 | 88.97 | 88.82/91.92 | 83.79 |
| MobRoFormer H & T | 10 | 113M | 84.61/84.73 | 91.36/88.35 | 90.54 | 91.90 | 56.74 | 77.91 | 88.51 | 88.34/91.67 | 83.76 |
| MöbiusBERT H & T | 11 | 104M | 84.49/84.49 | 91.59/88.59 | 91.21 | 92.09 | 56.84 | 76.75 | 89.23 | 88.24/91.5 | 83.82 |
| MöbiusBERT H& T Ortho | 11 | 104M | 84.36/85.33 | 91.35/88.22 | 91.10 | 92.43 | 56.20 | 77.76 | 89.37 | 87.65/90.99 | 83.85 |
| Overall (Möbius) | / | / | **84.58/85.33** | **91.59/88.59** | **91.21** | **92.43** | **56.84** | **77.91** | 89.37 | 88.82/91.92 | **84.17** |

Table 2: Results on the GLUE benchmark. First part of the table are the results on the original models, BERT and RoFormer, trained with our setup. Second part are our models. We also introduce overview rows with "Overall (others)" and "Overall (Möbius)" with the best performers accross the two model categories. Best performers are marked in bold in the overview rows and underlined in the individual models. MNLI, QNLI, SST-2 and RTE are measured using Accuracy, CoLA - Matthews correlation coefficient, STS-B - Spearman correlation coefficient, QQP and MRPC - F1 scores and Accuracy. Notation:
H: 50% Heads with MöbiusAttention, 50% vanilla;
T: Linear query Transformation before Möbius;
O: Orthogonal initialization strategy for the weights

**Results and Analysis**    Our models achieve superior performance to the baselines across several GLUE tasks, namely MNLI, QQP, QNLI, SST-2 and RTE as shown in Table 2. Both MöbiusBERT models also outperform their host model, BERT, in the MRPC task. However, none of the reviewed

models, including RoFormer, outperform BERT for the STS-B task. We also note that only one variant of our models outperforms BERT on SST-2, a sentiment classification task for movie reviews. Upon examining the two datasets, STS-B and SST-2, we found inconsistencies which we report in Appendix A.5. Due to the detected issues, we consider both training datasets to be of insufficient quality. To address this, we created a more challenging SST-2 version, details on which are provided in Appendix A.5.3. Our models outperform or match their host models on this more difficult SST-2 version, with scores reported in Table 6. No adjustments were conducted for STS-B as it is a regression task where labeling requires more resources and does not fall within the scope of this work.

**Analysis of MöbiusAttention**    To analyze the MöbiusAttention mechanism, we closely examine the learned Möbius weights and the resulting attention values. Our observations are as follows:

a) **Learned Geometries**: Our analysis of the learned Möbius weights reveals that the model captures a diverse range of complex geometries, as illustrated in Fig. 3. The heatmap displays the distribution of these geometries across different attention heads in the first and last layers of the model. Key observations include:

- **Layer-level Specialization**: The model decides for a layer-level geometry specialization by clearly favouring the circular and elliptic geometries in the last layer, but neglecting the circular one in the first layer.

- **Head-level Specialization**: The distribution varies on a head-level too, a sign for specialization of the heads. This is also evident in Fig. 6 and 7, examples of how geometries might change after finetuning on different tasks.

- **Beyond Circular Geometry**: Consistent with RoFormer's RoPe, the model emphasizes circular geometry, particularly in the last layer. However, it extends beyond the circular geometry supported by RoFormer, especially in the first layer, facilitating more complex reasoning.

b) **Learning to "Forget"**: The examination of the attention values obtained via MöbiusAttention show that vanilla attention and MöbiusAttention adopt different approaches to detecting important information. As seen in the attention heatmaps in Fig. 8 in the Appendix, vanilla attention almost never assigns zero attention score to a token pair. In contrast, MöbiusAttention gives most of the pairs zero score and only a few a non-zero one. Accordingly, instead of learning on what to "focus", MöbiusAttention learns what to "forget". The vanilla model has difficulty to give zero value to entires of the attention matrix due to using linear transformation on a group of tokens. Accordingly, it is hard for the model to forget elements, but Möbius can give a zero value to elements as the transformation can deform the distribution. As a result, the Möbius transformation is not limited to keep some irrelevant entries non-zero to keep the group similarities. Therefore, the mixture of Möbius and vanilla attention shows very promising results.

**Memory and Time Complexity**    MöbiusBERT demonstrates comparable pre-training efficiency to our BERT baseline, requiring the same pre-training duration of 26 hours. However, MöbiusBERT achieved this performance with a reduced memory footprint. Specifically, MöbiusBERT utilized 104 million parameters, whereas BERT required 110 million parameters. It is important to note that for MöbiusAttention, we counted the real and imaginary components of the complex-valued parameters separately, rather than combining them into a single parameter.

**Ablation Study**    To ensure maximal comparability with the BERT baseline, we adopted all hyperparameters used in the original BERT model without any further optimization or tuning (see Appendix A.3 and A.6 for details). Consequently, our ablation study focuses solely on variations within the MöbiusBERT architecture. Additionally, we maintain the same number of parameters by adjusting the number of layers.

We investigated the impact of Möbius attention placement within the transformer architecture. We experimented with four configurations: a) **Top Layer (10 Layers)**: A single Möbius attention layer positioned at the very beginning of the transformer stack, b) **Stacked Layers (9 Layers)**: Two consecutive Möbius attention layers at the beginning of the stack, c) **Framed Architecture (9 Layers)**: Möbius attention layers flanking the transformer stack (one at the beginning and one at the end), and d) **Alternating Layers (8 Layers)**: Three Möbius attention layers interspersed

Figure 3: Heatmap of geometry counts in different Möbius heads in 1st and last layers.
Model: MöbiusBERT H & T.
The count has been established through a visual inspection of the geometries.

throughout the stack, each separated by two vanilla attention layers. The number of layers in each configuration matches BERT's parameter size, accounting for the additional parameters introduced by MöbiusAttention.

Our findings, listed in Appendix A.7, revealed that both the stacked and alternating configurations yielded inferior performance compared to the framed model. This suggests potential overfitting within these architectures. Conversely, the framed architecture appears to introduce complexity in a controlled manner, mitigating overfitting. The initial Möbius attention captures intricate patterns, followed by vanilla attention layers that focus on specific aspects within those patterns and refine them. The final Möbius attention leverages the refined representation for even more complex reasoning.

## 6 CONCLUSION

In this paper, we introduce MöbiusAttention, a novel attention mechanism based on Möbius transformations. It offers greater expressiveness compared to traditional attention by leveraging Möbius transformations' unique capabilities. These transformations enable mappings between different geometries like line to line or circle, representing various shapes such as Circular, Hyperbolic, Loxodromic, Parabolic, and Elliptic. We observe that our models using MöbiusAttention learn not only the various geometries, but also different distributions of them in different parts of the model. The models also show clear preferences towards certain geometries in the distinct layers, exemplifying the benefits of an approach supporting big geometrical variability.

We integrated MöbiusAttention into BERT and RoFormer, forming MöbiusBERT and MobRoFormer, and evaluated their performance on GLUE benchmarks. Results show our models outperform their baseline on various tasks and on average with our MöbiusBERT models having fewer parameters (about 104M versus 110M) and no increase in training time. Our ablation study found that combining Möbius attention with traditional attention achieved the best performance among various architectural options. We specifically create a mixed-head model version where we allow for even more freedom in the model adjustments, allowing for tailoring to different use-cases.

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

## A APPENDIX

This section provides supplementary materials for our paper titled "Leveraging Möbius Transformations for Improved Attention in Transformer Models". It includes details on our motivation for the experimental setup, the architecture of MöbiusBERT, the hyperparameter settings used for pre-training, an introduction to the GLUE tasks, the hyperparameter specifications for GLUE fine-tuning, results from the ablation study, visualizations of standard attention mechanisms, and analysis of MöbiusAttention. We also provide a discussion on the limitations, A.9, and broader impact, A.10 of MöbiusAttention.

Our codebase is released on `https://anonymous.4open.science/r/MobiusAttention-4989/README.md`

### A.1 MOTIVATION OF THE EXPERIMENTAL SETUP

We integrate MöbiusAttention within the BERT architecture (Devlin et al., 2019) for several reasons. Firstly, BERT's widespread adoption and popularity make it a common benchmark for comparison in NLP tasks, e.g., Liu et al. (2019); Su et al. (2024); He et al. (2021). Secondly, BERT serves as the foundation for numerous high-performing models, such as RoBERTa (Liu et al., 2019), DeBERTa (He et al., 2021), Albert (Lan et al., 2019), etc. Finally, BERT's size allows for efficient training with limited resources compared to larger models that would require significantly longer training times (e.g., GPT3 (Brown et al., 2020), Mistral 7B (Jiang et al., 2023)). Our primary goal is to demonstrate performance improvements with MöbiusAttention without incurring substantial increases in memory and time complexity. As achieving state-of-the-art results often demands significantly more computational resources, BERT presents itself as the ideal candidate for our study.

Instead of the original BERT framework (Devlin et al., 2019), which was also followed in RoFormer (Su et al., 2024), we choose the newer MosaicBERT framework (Portes et al., 2023). BERT is originally trained on TPUs, while our hardware configuration consists of GPUs. The MosaicBERT framework offers training optimizations for BERT specifically designed for A100 GPUs, aligning perfectly with our setup. Additionally, the Toronto BookCorpus dataset (Zhu et al., 2015) used for pre-training BERT was not publicly available during our research timeframe. The authors of the MosaicBERT framework have accordingly chosen to work on the C4 dataset and provided the used codebase for full reproducibility of their BERT baseline, alleviating the dataset-mismatch challenge.

### A.2 ARCHITECTURE AND DESIGN

In this section we provide details on the architecture and design of our MöbiusAttention-enhanced models. Specifically, we present the mixed-head version as it achieves the highest performance. However, we also explain all alterations required for a layer with only MöbiusAttention heads.

Each block using MöbiusAttention is realized in Complex space. This causes several adjustments to the block which we build as follows:

Given our complex input $I = I_r + I_i \cdot i$, we first pass the values in the real and imaginary channels through a single LayerNorm instance ($LN_1$) before performing MöbiusAttention and vanilla attention (Eq. (10-12)). We add the real and imaginary parts of the MöbiusAttention output in order to obtain only one channel, allowing us to concatenate the outputs from the two types of attention heads (Eq. (13)). Next, we apply a linear layer (LL) and perform dropout, Eq. (14), add the residual connections, Eq. (15), and apply again LayerNorm ($LN_2$ instance), Eq. (16).

$$I' := I'_r + I'_i \cdot i = LN_1(I_r) + LN_1(I_i) \cdot i \tag{10}$$

$$A_r + A_i \cdot i = MobAtt(\mathcal{T}_q(I'), \mathcal{K}(I'), \mathcal{V}(I')) \tag{11}$$

$$A_{vanilla} = Att(Q_{vanilla}(I'_r + I'_i), K_{vanilla}(I'_r + I'_i), V_{vanilla}(I'_r + I'_i)) \tag{12}$$

$$A_{joined} = A_{vanilla} \| (A_r + A_i) \tag{13}$$

$$A' = Dropout(LL(A_{joined})) \tag{14}$$

$$A' + = I'_r \tag{15}$$

$$A' = LN_2(A'). \tag{16}$$

We then pass $A'$ through a Feed Forward Layer as defined in BERT ($FFL$), Eq. (17). Finally, we add again residual connections, Eq. (18), and apply a LayerNorm ($LN_3$) to get the output $O$.

$$A'' = FFL_r(A') \tag{17}$$

$$A'' + = A' \tag{18}$$

$$O = LN_3(A''). \tag{19}$$

Those adaptations are visually shown in Figure 4a. We note that we do not follow the Complex version of backpropagation (Benvenuto & Piazza, 1992) or use complex-valued normalization layers (Eilers & Jiang, 2023; Trabelsi et al., 2018) for computational efficiency.

The remaining blocks (excluding the first and last) are standard Transformer blocks.

We note that in our mixed-head models we add the real and imaginary parts of the MöbiusAttention output before the application of the linear layer, as shown in Eq. (13). This is required to obtain only one channel, allowing to concatenate the outputs from the two types of attention heads. For architectures with layers using MöbiusAttention heads, it is possible to add the two channels at the end of the block. Benefits of this approach are that we can apply two linear projections separately on the channels, as well as separate feed forward blocks, giving the model additional freedom, yet, at the expense of additional parameters. This is the approach we adopted for the models with no head division. The described alternative follows Eq. (20)-(27).

$$I' := I'_r + I'_i \cdot i = LN_1(I_r) + LN_1(I_i) \cdot i \tag{20}$$

$$A_r + A_i \cdot i = Att(\mathcal{T}_q(I'), \mathcal{K}(I'), \mathcal{V}(I')) \tag{21}$$

$$A'_r, A'_i = Dropout(LL_r(A_r), LL_i(A_i)) \tag{22}$$

$$A'_r + = I'_r, A'_i + = I'_i \tag{23}$$

$$A'_r = LN_2(A'_r), A'_i = LN_2(A'_i) \tag{24}$$

$$A''_r = FFL_r(A'_r), A''_i = FFL_i(A'_i) \tag{25}$$

$$A''_r + = A'_r, A''_i + = A'_i \tag{26}$$

$$O_r = LN_3(A''_r), O_i = LN_3(A''_i). \tag{27}$$

### A.3 HYPERPARAMETERS CHOICE

The training configuration for the models is specified with various hyperparameters to optimize performance. We follow completely the hyperparameters set in the MosaicBERT BERT-Base framework Portes et al. (2023), which are also chosen in adherence to the ones in the original BERT framework Devlin et al. (2019). The maximum sequence length (`max_seq_len`) is set to 128, and the tokenizer used is `bert-base-uncased` from Hugging Face. The masking probability (`mlm_probability`) is configured at 0.15 as originally done in BERT. All models have 12 attention heads with varying number of layers to ensure comparable model sizes. The maximum position embedding is set to 512 and an attention dropout probability of 0.1.

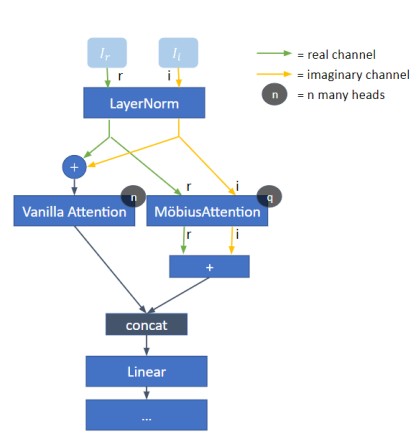

(a) Structure of a block using mixed attention heads.

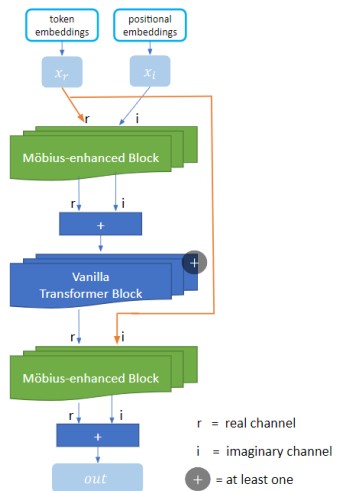

(b) Architecture of MöbiusBERT.

Figure 4: Architecture of MöbiusBERT and the blocks using MöbiusAttention.

| Hyperparameter | Value |
|---|---|
| Max Sequence Length | 128 |
| Tokenizer Name | bert-base-uncased |
| MLM Probability | 0.15 |
| Number of Attention Heads | 12 |
| Number of Hidden Layers | Varies for different models |
| Max Position Embedding | 512 |
| Attention Dropout Probability | 0.1 |
| Learning Rate | $5.0 \times 10^{-4}$ |
| AdamW Betas | (0.9, 0.98) |
| AdamW Epsilon | $1.0 \times 10^{-6}$ |
| Weight Decay | $1.0 \times 10^{-5}$ |
| Warmup Duration | 6% of training |
| LR Decay Factor | 0.02 |
| Max Training Samples | 275M |
| Evaluation Interval | 2000 batches |
| Global Train Batch Size | 4096 |
| Seed | 17 |
| Device Eval Batch Size | 128 |
| Device Train Microbatch Size | 128 |
| Precision | AMP BF16 |

Table 3: Hyperparameter Settings for Pre-Training

The training data loader uses 8 workers, shuffles the data, and drops the last incomplete batch. The evaluation data loader also employs 8 workers but does not shuffle the data and retains the last batch. The optimization process utilizes a `linear_decay_with_warmup` scheduler, with a warmup period covering 6% of the total training duration, and decays the learning rate to 0.02 times the initial rate. The optimizer is `decoupled_adamw` with a peak learning rate of $5.0 \times 10^{-4}$, beta values of (0.9, 0.98), epsilon of $1.0 \times 10^{-6}$, and a weight decay of $1.0 \times 10^{-5}$.

The global training batch size is set to 4096, and the random seed for reproducibility is 17. Both the evaluation and training microbatch sizes are 128, and mixed precision training is enabled with AMP BF16.

Evaluations are conducted every 2000 batches model checkpointing every 3500 batches. The overall training spans 70,000 steps, providing sufficient iterations to fine-tune the model parameters and achieve the desired accuracy.

## A.4 GLUE BENCHMARK

The General Language Understanding Evaluation (GLUE) benchmark (Wang et al., 2018) is a collection of diverse natural language understanding tasks designed to evaluate and analyze the performance of models across a range of linguistic phenomena. In our experiments, we use the following GLUE tasks:

- **MNLI (Multi-Genre Natural Language Inference) (Williams et al., 2017):** This task requires the model to classify pairs of sentences into three categories: entailment, contradiction, or neutral. It evaluates the model's ability to understand relationships between sentences across multiple genres. MNLI is further divided into two subsets:
  - **MNLI-m (Matched):** The evaluation set contains examples that are from the same genres as the training data.
  - **MNLI-mm (Mismatched):** Mismatched version where the evaluation set contains examples from different genres than those found in the training data, testing the model's robustness across varying contexts.

  We evaluate MNLI using Accuracy.

- **QQP (Quora Question Pairs 2) (Iyer et al., 2017):** The goal is to determine if two questions from the Quora website are semantically equivalent. This task tests the model's ability to identify paraphrased questions. For evaluation we use both Accuracy and F1 Score, since both metrics are used in different papers, e.g., Devlin et al. (2019); Qin et al. (2022).

- **QNLI (Question Natural Language Inference) (Rajpurkar et al., 2016):** Based on the Stanford Question Answering Dataset (SQuAD), this task involves determining whether a context sentence contains the answer to a given question, formulated as a binary classification problem. The chosen evaluation metric is Accuracy.

- **SST-2 (Stanford Sentiment Treebank) (Socher et al., 2013):** A sentiment analysis task where the model predicts the sentiment (positive or negative) of a given sentence. This evaluates the model's ability to understand and classify emotional content. We evaluate using Accuracy.

- **MRPC (Microsoft Research Paraphrase Corpus) (Dolan & Brockett, 2005):** This task involves identifying whether pairs of sentences are paraphrases of each other. It assesses the model's capability to recognize semantic similarity. For MRPC we provide results both for F1 Score and for Accuracy.

- **STS-B (Semantic Textual Similarity Benchmark) (Cer et al., 2017):** The model must predict the similarity score between two sentences on a scale from 0 to 5. This task measures the model's ability to capture the degree of semantic equivalence. We use the Spearman correlation coefficient as the evaluation metric.

- **CoLA (Corpus of Linguistic Acceptability) (Warstadt et al., 2019):** The objective is to determine whether a given sentence is grammatically acceptable. This task tests the model's understanding of linguistic acceptability. All results we provide for CoLA measure the Matthews correlation coefficient.

- **RTE (Recognizing Textual Entailment) (Dagan et al., 2005; Giampiccolo et al., 2007; Bentivogli et al., 2009):** The model must classify pairs of sentences as entailment or not entailment. This task is similar to MNLI but involves data from a variety of sources with fewer training examples. The results we provide for RTE show the accuracy.

We note that we explicitly do not fine-tune on the WNLI (Winograd NLI) task (Levesque et al., 2012) as it considered to be challenging to work with (Liu et al., 2019; Devlin et al., 2019; Portes et al., 2023), leading to its omission in the benchmarks of multiple models, e.g., BERT (Devlin et al., 2019), RoFormer (Su et al., 2024).

## A.5 GLUE LIMITATIONS

In our study, we identified several issues with the STS-B and SST-2 benchmarks that can impact the performance and reliability of models evaluated on these datasets. In the following subsections, we detail the inconsistencies observed in the STS-B benchmark and the issues found in the SST-2 dataset. Additionally, we describe our methodology for improving the SST-2 benchmark and present the results achieved with the revised dataset.

### A.5.1 INCONSISTENCIES IN THE STS-B BENCHMARK

Our analysis of the STS-B benchmark revealed several inconsistencies affecting the dataset's quality. The STS-B benchmark is a regression task where models predict sentence similarity scores on a scale from 1 to 5. However, we found that these similarity scores often exhibit significant deviations. Specifically:

- Table 4 illustrates instances where identical sentence pairs are assigned similarity scores with deviations of up to 0.6.
- Table 5 highlights cases where only the subject of the sentence differs within the pair, yet the similarity scores vary significantly accross the pairs.

These examples underscore inconsistencies in labeling within the STS-B benchmark, although a comprehensive review of all anomalies is beyond the scope of this work.

| Id | Sentence 1 | Sentence 2 | Score |
|---|---|---|---|
| 2010 | Imagine a place that's % white and % black. | Imagine a place with % men and % women. | 1.00 |
| 2160 | Imagine a place with % men and % women. | Imagine a place that's % white and % black. | 1.60 |
| 202 | I don't prefix or suffix everything with "you (...) | You should stop prefixing or suffixing everyth(...) | 3.00 |
| 23709 | You should stop prefixing or suffixing everyth(...) | I don't prefix or suffix everything with "you (...) | 2.40 |
| 2056 | Ah ha, ha, ha, ha, ha! | Ha, ha, ha, ha, ha, ha! | 4.50 |
| 2367 | Ha, ha, ha, ha, ha, ha! | Ah ha, ha, ha, ha, ha! | 5.00 |
| 121 | A man is playing a piano. | A man is playing a flute. | 2.00 |
| 122 | A man is playing a flute. | A man is playing a piano. | 1.60 |
| 203 | A rooster pecks at a dead mouse. | A chicken is pecking at a dead mouse. | 4.00 |
| 704 | A chicken is pecking at a dead mouse. | A rooster pecks at a dead mouse. | 3.60 |
| 519 | A man is playing a guitar. | A man is playing a flute. | 1.583 |
| 518 | A man is playing a flute. | A man is playing a guitar. | 2.00 |

Table 4: Examples for STS-B Scores Inconsistencies

| Id | Sentence 1 | Sentence 2 | Score |
|---|---|---|---|
| 55 | A man is playing the **guitar**. | A man is playing the **drums**. | 1.56 |
| 520 | A man is playing a **piano**. | A man is playing a **guitar**. | 1.778 |
| 73 | A man is playing the **piano**. | A man is playing the **trumpet**. | 1.60 |
| 518 | A man is playing a **flute**. | A man is playing a **guitar**. | 2.00 |

Table 5: Examples for STS-B scores inconsistencies: all sentence pairs above are essentially the same - a man is playing a musical instrument which is different in the two sentences. Yet, each pair has a different score, ranging from 1.56 to 2.0

### A.5.2 ISSUES WITH THE SST-2 BENCHMARK

Our examination of the SST-2 dataset uncovered several significant issues:

- **Duplicate Instances**: We identified 8,727 duplicates out of 67,300 instances, constituting approximately 13% of the dataset.
- **Substring Overlaps**: There are 34,768 instances that are substrings of each other and have a word count > 5, such as "A B C D E" and "A B C D E F". Here we focus only on sentences with word count above 5 as shorter sequences are likely to occur often in other instances.
- **Train-Val Distribution Mismatch**: We found 10,663 instances of length 1 or 2 (around 15.8%), whereas the validation split contains less than 1% of such short instances.

These issues suggest that models might rely on memorization of short phrases rather than learning sentiment relationships effectively.

### A.5.3 AN IMPROVED SST-2 DATASET

To address these concerns, we created an improved version of the SST-2 dataset with the following modifications:

- Replaced 464 positive and 475 negative short reviews with longer, manually crafted reviews. We varied the writing styles and the genrse of the reviewed movies for maximum variability.
- Removed all duplicate instances and reduced overlapping substrings by retaining the longest instance or a "middle length" instance from groups with the same label, and preserving all instances from groups with varying labels.

While these modifications do not address all issues with the SST-2 dataset, they serve as a proof of concept for creating a more challenging and representative benchmark. The results obtained with this revised version are presented in Table 6.

| Model | Accuracy (%) |
|---|---|
| BERT | 91.63 |
| MobiusBERT H, T Ortho | 92.05 |
| MobiusBERT H & T | 91.78 |
| RoFormer | 92.09 |
| RoFormer H & T | 92.24 |
| RoFormer H | 92.09 |

Table 6: Results on the adapted SST-2 dataset.

### A.6 HYPERPARAMETERS CHOICE FOR GLUE FINE-TUNING

The fine-tuning process for the BERT model on the GLUE benchmark is again configured in adherence to the MosaicBERT framework we follow Portes et al. (2023). It leverages multiple GPUs by running the various GLUE tasks in parallel. The random seed for reproducibility is set to 19, and the precision is configured to `bf16`.

We note that the fine-tuning for the tasks RTE, MRPC and STSB is based on a MNLI-tuned checkpoint, as suggested by the MosaicBERT framework and in adherence to to Izsak et al. (2021) on efficient BERT training.

| Hyperparameter | Value |
|---|---|
| Parallel Execution | true |
| Default Seed | 19 |
| Precision | AMP BF16 |
| Tokenizer Name | bert-base-uncased |

Table 7: General settings for fine-tuning on GLUE

The training scheduler is configured with a linear decay with warmup, with the warmup duration set to 6% of the training duration and the final learning rate factor (`alpha_f`) set to 0.

| Scheduler | Value |
|---|---|
| Name | linear_decay_with_warmup |
| Warmup Duration | 0.06 of training |
| Final Learning Rate Factor | 0.0 |

Table 8: Scheduler settings

Each GLUE task has specific configurations, including the number of random seeds and the number of checkpoints to retain. For example, MNLI retains one checkpoint locally, while other tasks do not retain any checkpoints.

| Task | Seeds | Checkpoints to Keep |
|---|---|---|
| MNLI | - | 1 |
| RTE | [19, 8364, 717, 10536, 90166] | 0 |
| QQP | - | 0 |
| QNLI | - | 0 |
| SST-2 | [19, 8364, 717] | 0 |
| STS-B | [19, 8364, 717, 10536, 90166] | 0 |
| MRPC | [19, 8364, 717, 10536, 90166] | 0 |
| CoLA | [19, 8364, 717, 10536] | 0 |

Table 9: Task-specific settings for GLUE fine-tuning

## A.7 ABLATION EXPERIMENTS

The table below presents the results of our ablation study on Möbius attention placement within the BERT architecture. We evaluated four configurations:

- **Top Layer:** A single Möbius attention layer at the beginning.
- **Stacked Layers:** Two consecutive Möbius attention layers at the beginning.
- **Framed Architecture:** Möbius attention layers at both the beginning and the end.
- **Alternating Layers:** Three Möbius attention layers interspersed throughout.

We note that we choose the number of layers to ensure that each configuration maintains a comparable parameter size to the BERT model, 110 million parameters.

| Model | Layers | Parameters | MNLI-(m/mm) | QQP (Acc/F1) | QNLI | SST-2 | CoLA | RTE | STS-B | MRPC (Acc/F1) |
|---|---|---|---|---|---|---|---|---|---|---|
| BERT (our baseline) | 12 | 110M | **84.46/85.14** | 91.23/88.13 | 90.65 | 92.16 | **56.29** | 76.61 | **89.79** | 87.40/90.88 |
| Möbius Framed | 9 | 105M | 83.77/84.73 | **91.37/88.30** | **91.05** | **92.20** | 54.00 | **76.97** | 89.01 | 88.29/91.53 |
| Möbius Stacked | 9 | 105M | 83.78/84.42 | 91.19/88.08 | 89.51 | 92.09 | 52.90 | 73.65 | 89.14 | **88.77/92.09** |
| Möbius Alternating | 8 | 106M | 81.45/82.33 | 90.74/87.46 | 88.17 | 91.48 | 45.69 | 73.50 | 88.24 | 86.32/90.15 |
| Möbius Top | 10 | 104M | 84.04/84.42 | 91.26/88.22 | 90.26 | 92.28 | 52.10 | 72.78 | 88.69 | 87.01/90.53 |

Table 10: Results on the GLUE benchmark. Best performers are marked in bold.

## A.8 ANALYSIS OF MÖBIUSATTENTION

In this section we provide in-depth analysis of the weights learned using MöbiusAttention, of their properties, and of the resulting attention outputs. Of particular interest for us are the learned Möbius transformation parameters due to them being responsible for the learned geometries. Accordingly, we offer more details on them in Section A.8. Then, in Section 5, we present a comparison of the attention outputs obtained through vanilla attention and MöbiusAttention.

Delving deeper into the inner workings of MöbiusBERT, we specifically examined the weights learned for the query element, which undergoes the Möbius transformation. This analysis revealed

that the model can learn weights that exhibit various geometric patterns. These patterns include Loxodromic, Elliptic, Hyperbolic, Parabolic, and Circular geometries, as visualized in Figure 7. This diversity in weight geometry suggests that MöbiusAttention is not restricted to a single mode of operation but can adapt its behavior based on the specific context and task at hand.

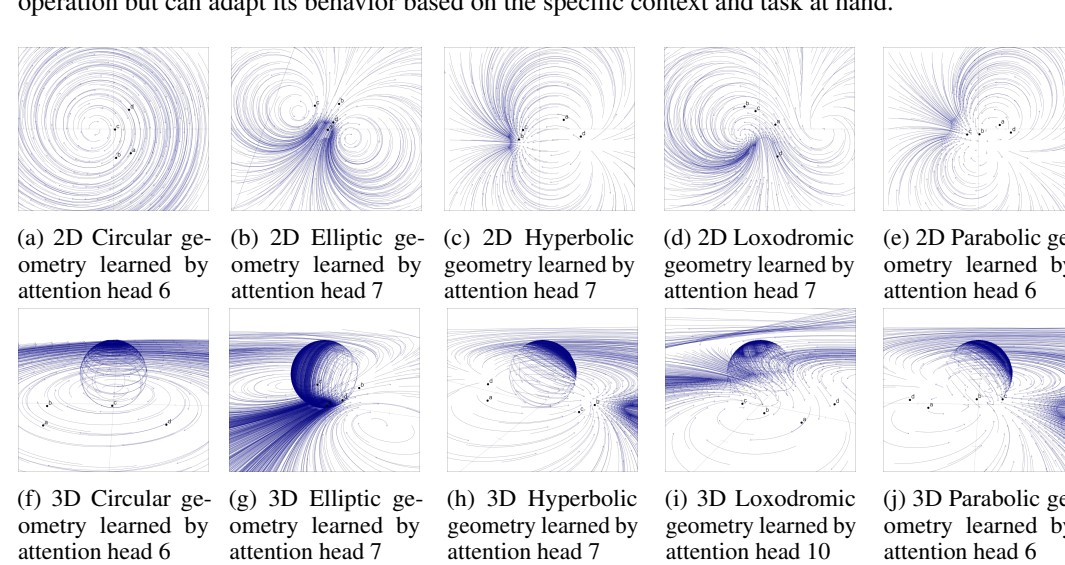

(a) 2D Circular geometry learned by attention head 6

(b) 2D Elliptic geometry learned by attention head 7

(c) 2D Hyperbolic geometry learned by attention head 7

(d) 2D Loxodromic geometry learned by attention head 7

(e) 2D Parabolic geometry learned by attention head 6

(f) 3D Circular geometry learned by attention head 6

(g) 3D Elliptic geometry learned by attention head 7

(h) 3D Hyperbolic geometry learned by attention head 7

(i) 3D Loxodromic geometry learned by attention head 10

(j) 3D Parabolic geometry learned by attention head 6

Figure 5: Various Möbius transformations learned by MöbiusBERT in the first and the last layers using MöbiusAttention. Visualizations are created using the visualization tool `https://timhutton.github.io/mobius-transforms/` to get our visualizations.

Additionally, MöbiusAttention shows the ability to adapt to different tasks via different geometries. To showcase this we refer to the following visualizations:

- Fig. 6 and 7 are examples of how geometries might change after finetuning on different tasks. Fig. 7 shows in the first row how the geometry in dimension 17 in the 6th Mobius head, layer 1 transforms after different tasks. The second row shows the same for the geometry in dimension 16. We can see that each task has led to a different geometry. Fig. 6 shows an example from a different head, proving that this is not an occurrence only in one head.

- We also point to the relationship between Fig. 6b and Fig. 7e, both of which depict results after finetuning on the same task (SST2). Even though in different heads, both geometries have evolved in the hyperbolic direction.

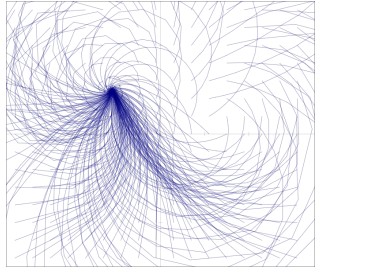   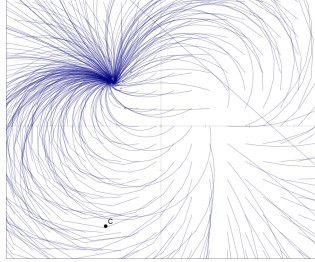

(a) Geometry **before** finetuning.          (b) Geometry **after** finetuning on SST2 task.

Figure 6: Geometry for dimension 1 in head 0, layer 0 before and after finetuning for the SST2 task.

**Attention Heatmaps**  We have also examined the outputs of the two attention mechanisms, MöbiusAttention and vanilla. In Fig. 8 we provide the heatmap visualizations for the attention mechanism over different attention heads from our MöbiusBERT model (H & T version, i.e. 6 heads MöbiusAttention, 6 heads vanilla, 11 layers, linear-layer query before Möbius). It can be observed

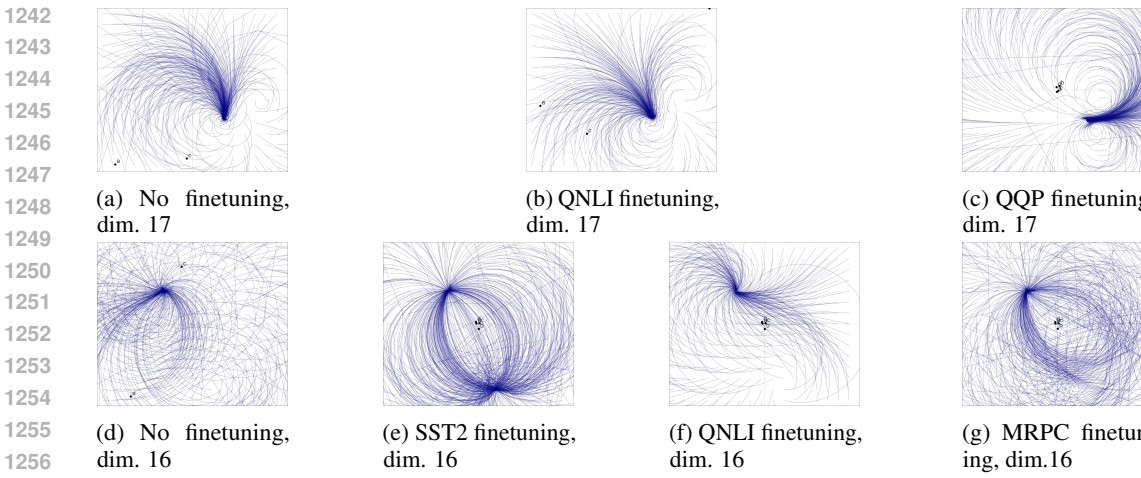

(a) No finetuning, dim. 17

(b) QNLI finetuning, dim. 17

(c) QQP finetuning, dim. 17

(d) No finetuning, dim. 16

(e) SST2 finetuning, dim. 16

(f) QNLI finetuning, dim. 16

(g) MRPC finetuning, dim.16

Figure 7: Möbius transformations showing how the learned geometry might (or might not) change through fine-tuning to adapt better. Images (a)-(f) are examples where a non-discernible geometry type was moved towards a discernible one during finetuning. Images (d), (g), however, are proof that such non-discernible geometries are not always further refined. The visualizations here all from the first layer head twelve, i.e. the sixth head of those using MöbiusAttention. "Dim." represents the dimension.

that vanilla attention rarely assigns a zero attention score to a token pair, while MöbiusAttention does this often. We have discussed this phenomenon in Section 5.

## A.9 LIMITATIONS

While our work on MöbiusAttention shows that it can lead to performance improvements accross various tasks, it also presents several potential limitations to consider:

- **Overfitting:** As demonstrated in the ablation study detailed in Section 5 and Appendix A.7, MöbiusAttention can be susceptible to overfitting in certain architectures. Careful design choices, such as placement and number of MöbiusAttention layers, are crucial to mitigate this risk.

- **Linear Relationships and Generalizability:** Our approach might exhibit inferior performance on tasks that rely heavily on rules with linear relationships or require broad generalization across diverse categories. This is discussed further in Section 5. For such tasks, simpler attention mechanisms might be more effective.

- **Computational Complexity:** MöbiusAttention operates in the complex domain and utilizes Möbius transformations for the query, requiring eight parameters per attention head. Compared to vanilla attention, this leads to increased computational complexity. To address this, we reduced the depth of models integrating MöbiusAttention compared to baseline models. However, our results (Section 5) show that even with fewer layers, MöbiusAttention leads to performance improvements on many tasks.

- **Limited Evaluation Scope:** Currently, MöbiusAttention has only been evaluated on GLUE tasks. Future work should explore its effectiveness on a broader range of NLP tasks, such as machine translation. Additionally, the potential applicability of MöbiusAttention in computer vision tasks remains untested and is a promising avenue for further investigation.

## A.10 BROADER IMPACT

Transformer-based models with enhanced attention mechanisms, like MöbiusAttention, hold significant promise for boosting performance across diverse fields such as NLP, computer vision, and signal processing. These advancements can lead to more accurate and efficient models, impacting

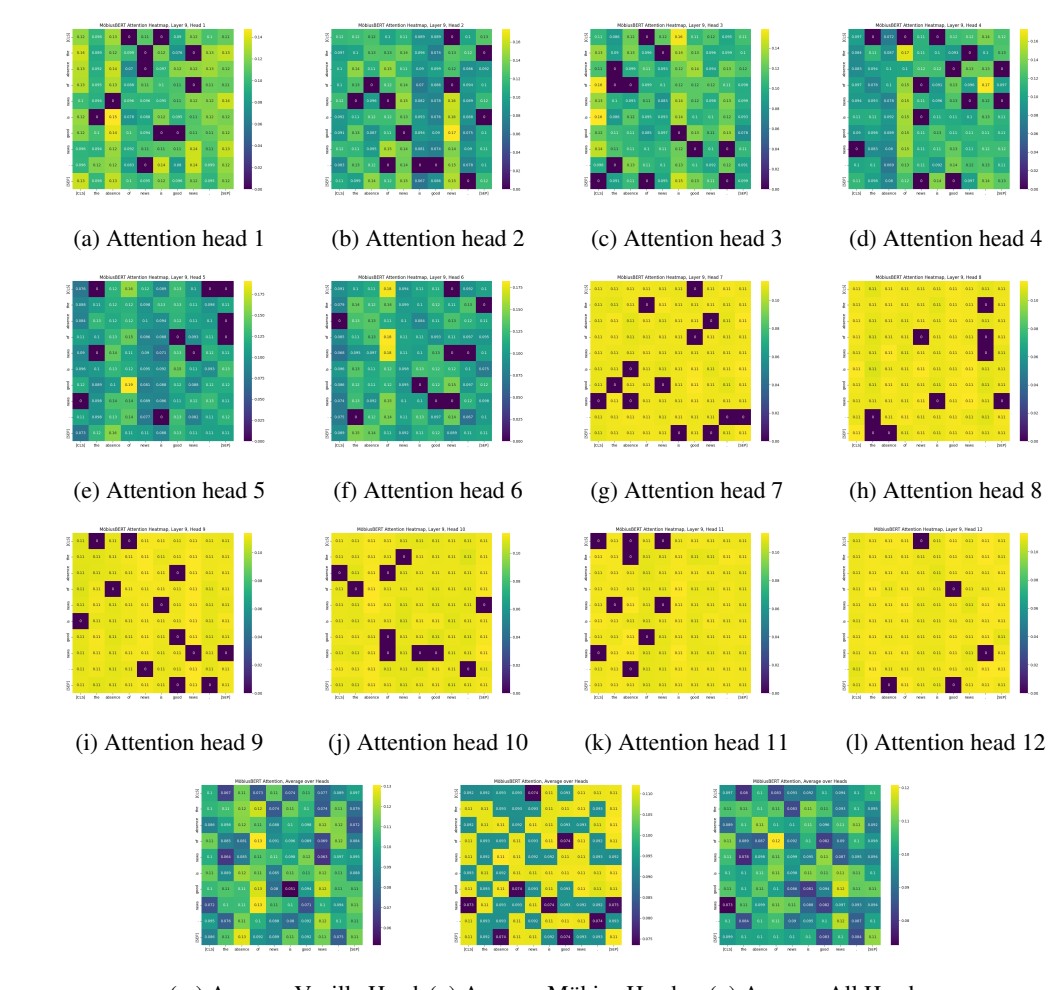

(a) Attention head 1    (b) Attention head 2    (c) Attention head 3    (d) Attention head 4

(e) Attention head 5    (f) Attention head 6    (g) Attention head 7    (h) Attention head 8

(i) Attention head 9    (j) Attention head 10    (k) Attention head 11    (l) Attention head 12

(m) Average Vanilla Heads (n) Average Möbius Heads    (o) Average All Heads

Figure 8: Heatmaps over attention from the last layer, layer 12, of our MöbiusBERT (H & T). Heads 1 to 6 are vanilla attention, 7 to 12 - MöbiusAttention.

applications like machine translation, object detection/classification, or conversational AI. Successfully tackling these tasks unlocks a range of benefits, including easier cross-cultural communication, improved accessibility for people with sensory disabilities, educational enhancements, and new technological innovations.

However, alongside these benefits, there are significant risks associated with the misuse of these powerful models, e.g.:

- **Disinformation and Misinformation:** Enhanced Transformer-based models can be used to generate highly convincing text or images, which may be utilized for spreading false information deliberately, thus, amplifing the reach and impact of fake news.

- **Deepfake Text Generation:** The ability to produce human-like text can be exploited to create deepfake content, such as fabricated emails, articles, or social media posts. This can lead to identity theft, fraud, and other malicious activities that undermine trust in digital communications.

- **Bias and discrimination:** Despite improvements in attention mechanisms, Transformer models can still perpetuate and even amplify biases present in training data. Misuse in automated decision-making processes (e.g., hiring, law enforcement) can lead to discriminatory practices and reinforce existing social inequalities.

Those examples stress the need for adopting vigilance in data curation, model development practices that minimize bias amplification, and fostering public awareness of the capabilities and limitations of AI-generated content.

We also note that training large AI models can be computationally expensive, leading to a significant carbon footprint. Researchers and developers should strive for energy-efficient training methods and utilize renewable energy sources whenever possible. Responsible hardware choices and model optimization techniques can further minimize the environmental impact of MöbiusAttention-based applications.

By discussing these potential issues, we hope to contribute to a responsible utilization of MöbiusAttention, maximizing its positive impact on society and the environment.

