# OpenReview forum: "Expanding Expressivity in Transformer Models with MöbiusAttention"
_ICLR.cc/2025/Conference — ICLR 2025 Conference Withdrawn Submission_

### Official Review · Reviewer_z7FV · 2024-11-02

**Soundness:** 2
**Presentation:** 4
**Contribution:** 2
**Rating:** 3
**Confidence:** 3

**Summary:**

This paper proposes a new novel class of attention mechanism, which is MöbiusAttention that is based on geometric backgrounds. MöbiusAttention develops the standard linear attention mechanism with Möbius transformation and complex domain extension. In addition to provide the theoretic understanding of MöbiusAttention, this paper applies it to BERT and RoFormer, then conducts the benchmark GLUE task.

**Strengths:**

- Relationship between geometry and attention mechanism is an underexplored research field. This paper explores this field based on projective geometry.

- Strong mathematical background: The proposed methodology, MöbiusAttention, is motivated by rigorous mathematical background. It brings reliability for the claim of geometric characteristics embedded in MöbiusAttention mechanism different from the standard attention mechanism. In addition, the extension to complex domain of token representation is smoothly related to the proposed methodology.

- Well-written texts: Even though several claims seem questionable, at least they were easy to understand because of the clear writing.

**Weaknesses:**

- The paper’s main argument is not persuasive. This paper argues MöbiusAttention can enhance expressivity based on its non-linear operation, Möbius transformation. However, I guess Möbius transformation would not bring significant expressivity enhancement because it is constrained to be projective linear group (PGL) which is a scaled general linear group.
- Experiment results do not support the expressivity argument. First, despite of the enhanced expressivity (in hypothesis), the performance improvement in the GLUE task is not superior (around 0.2 point improvement on average). Second, the argument of overfitting problem due to the enhanced expressivity (in the paragraph, lines 360~370) needs more clarification.

- “Learning to Forget”: the argument “Fig. 8 in the Appendix, vanilla attention almost never assigns zero attention score to a token pair. In contrast, MöbiusAttention gives most of the pairs zero score and only a few a non-zero one” seems not supported by the results. In Figure 8, the average number of zero elements in Vanilla head (a~f) is 11.17, while Möbius head (g~l) is 9.67. More importantly, the attention matrix of MöbiusAttention seems almost uniform.
- “Memory and Time Complexity”: the comparison of parameters (104M vs 110M) seems negligible.

- No experimental comparison between other comparable works. The previous works, such as RoPE and NeuralAttention are not compared with MöbiusAttention.

Minor:
- In ‘Geometric Interpretation’ section, the orientation could be changed in ‘det M_{qj} = 1’ case, since orientation can be changed without changing the volume.
- In lines 314~317 “Möbius transformations offer a robust framework for analyzing and interpreting text”, this argument seems not verified by experiments. The authors might want to view the geometric shapes of some semantically similar text tokens.
- Even though this paper argues that MöbiusAttention does not increase computational complexity significantly, it is not experimentally compared. Due to the separated operations (layernorm, attention) for real and complex parts, it might increase computational cost.
- In Table 2, the ‘Overall’ scores seem not in fair comparison because the number of candidates that contribute to the maximum case selection are different, 2 candidates for baselines and 4 candidates for Möbius.

**Questions:**

- In Equations 7 & 8, I understood ‘w_{kj}’ and ‘w_{vj}’ are learnable weight parameters. However, they use the same notation with word embedding vector ‘w_i’ in the paragraph (lines 255~259).
- In Equation 9, I think ‘Q_ij=T_{qj}(rho_{ij})’ should be ‘Q_ij=M_{qj}(rho_{ij})’ because the output of T_{qj}(rho_{ij}) is 2-dim, so that it is unable to matrix-multiply Q (2xd dimension) and K (d dimension). Also, if ‘T_{qj}(rho_{ij})’ is correct, then it should be T_{qj}(rho^{h}_{ij}) because this map utilizes M which 2x2 matrix. Please clarify if I am wrong.
- Some table captions include weird newlines.

---

### Official Review · Reviewer_QQAq · 2024-11-02

**Soundness:** 2
**Presentation:** 2
**Contribution:** 2
**Rating:** 3
**Confidence:** 3

**Summary:**

The authors introduce MöbiusAttention, an attention mechanism that incorporates Möbius transformations within the query computation of Transformer models. Möbius transformations, known for their ability to map complex geometries, are utilized to enable Transformers to capture more intricate inter-token relationships beyond the capabilities of linear operations. The paper outlines the mathematical underpinnings of projective geometry and Möbius transformations, detailing how these concepts are integrated into the attention mechanism.  Experimental results demonstrate that MöbiusAttention-enhanced BERT achieves similar performance to the original BERT.

**Strengths:**

The integration of Möbius transformations into the attention mechanism is a novel approach that leverages complex geometric mappings.

**Weaknesses:**

1. The paper's motivation lacks rigorous justification and empirical support. Specifically, in the second paragraph of the introduction, the authors make several unsupported claims about the limitations of linear transformations and softmax in Transformer attention mechanisms:

    a) The authors assert that *"predominantly linear operations restrict the ability of models to capture complex linguistic dependencies,"* but provide no theoretical analysis or experimental evidence demonstrating these supposed limitations.

    b) The authors claim that linear operations lead to *"potential information loss within each attention layer"*. However, they don't elaborate on what type of information is lost or why this loss occurs.

2.  The experimental results raise significant concerns about the practical value of the proposed MöbiusAttention mechanism. While the authors introduce a mathematically sophisticated approach using Möbius transformations, the empirical results show that MöbiusAttention-based BERT performs comparably to the original BERT model, with no significant improvements in performance.

**Questions:**

Please address the weaknesses above.

---

### Official Review · Reviewer_QJec · 2024-11-03

**Soundness:** 3
**Presentation:** 2
**Contribution:** 2
**Rating:** 5
**Confidence:** 3

**Summary:**

This paper focuses on introducing non-linearity into attention mechanism, specifically exploring transformations capable of operating across diverse geometric spaces. The authors propose MÖBIUSATTENTION where the non-linearity is introduced through Möbius transformations, which can map points between different geometries, such as from a line to a circle, a circle to a line, and similarly among lines and circles, allowing for the model to capture more complex inter-token dependencies than traditional linear methods. The authors states that integrating MöbiusAttention can lead to improved performance across various NLP tasks without necessarily increasing the model size.

**Strengths:**

- The idea that introducing non-linearity into attention through Möbius transformations is interesting.
- The writing in first half part of paper is good and easy to follow. It gives enough backgrounds and a lot of details about the method.

**Weaknesses:**

- **Motivation is not clear.** I understand the intuition that Möbius transformations can capture more complex patterns and therefore improve the attention. However, I am not sure why we need to capture these patterns and what kind of tasks require these patterns.
- Some statements are inaccurate. For example:
  -  line 91: "integrating MöbiusAttention can lead to improved performance across various NLP tasks without necessarily increasing the model size."
  - line 1285: "However, our results (Section 5) show that even with fewer layers, MöbiusAttention leads to performance improvements on many tasks."
  - line 460: "As seen in the attention heatmaps in Fig. 8 in the Appendix, vanilla attention almost never assigns zero attention score to a token pair. In contrast, MöbiusAttention gives most of the pairs zero score and only a few a non-zero one."
  - These statements can not be drawn from experimental results.
- The experiments also have several limitations as follows:
  - **Performance is weak**: The performance improvements brought by MÖBIUSATTENTION are marginal.  For example, from Table 2, we can see MobRoFormer H & T can only achieve 0.27 performance improvement compared to RoFormer but consuming more parameters. Given this improvement is marginal, It is not clear that whether the performance improvement is significant. Therefore, I recommend the authors to test this.
  - **Lack of Efficiency Analysis**: Intuitively, enabling attention to capture more complex patterns will lead to an increase in FLOPs and inference latency. However, the authors do not provide analysis in these aspects.
  - **Lack of Large Scale Experiments**: Currently, the authors only mainly focused on small models and GLEU tasks. It is necessary to verify the effectiveness of MöbiusAttention in models with larger sizes and broader tasks.

**Questions:**

See above.

---

### Official Review · Reviewer_8fGZ · 2024-11-07

**Soundness:** 2
**Presentation:** 3
**Contribution:** 2
**Rating:** 3
**Confidence:** 4

**Summary:**

The paper replaces the self-attention mechanism in BERT with Möbius Attention, which is motivated by projective geometry.

The authors hypothesize and the linear projections of the query, keys, and values limit the representation ability of transformers. Moreover, they have a more targeted treatment of the position embeddings.

They implement a new attention mechanism and run ablations using the GLUE benchmark.

**Strengths:**

The method is pretty simple to understand and has interesting motivation from projective geometry.

**Weaknesses:**

The results just don't seem to be particularly strong. The level of improvement seen could just be from doing lots of ablations and model retrains with different seeds.

**Questions:**

The new attention mechanism doesn't seem to improve the GLUE benchmark very much. Is there a task where the authors think there might be a larger improvement? Why?

---

### Note · Authors · 2024-11-15

I have read and agree with the venue's withdrawal policy on behalf of myself and my co-authors.